# Effectiveness of Text Messaging as an Incentive to Maintain Physical Activity after Cardiac Rehabilitation: A Randomized Controlled Pilot Study

**DOI:** 10.3390/ijerph18126645

**Published:** 2021-06-21

**Authors:** Giulia Foccardi, Marco Vecchiato, Daniel Neunhaeuserer, Michele Mezzaro, Giulia Quinto, Francesca Battista, Federica Duregon, Roberto Carlon, Andrea Ermolao

**Affiliations:** 1Sports and Exercise Medicine Division, Department of Medicine, University of Padova, 35128 Padova, Italy; giulia.foccardi@gmail.com (G.F.); marcovecchiato.md@gmail.com (M.V.); mezzmiki@gmail.com (M.M.); giulia.quinto9@gmail.com (G.Q.); francesca.battista@studenti.unipd.it (F.B.); federica.duregon@unipd.it (F.D.); andrea.ermolao@unipd.it (A.E.); 2Clinical Network of Sports and Exercise Medicine of the Veneto Region, 35128 Padova, Italy; 3UOA Cardiology, Rehabilitation Cardiology, ASSL 6 “Euganea”, Cittadella Hospital, 35013 Cittadella, Italy; carlon.roberto@gmail.com

**Keywords:** ischemic cardiomyopathy, coronary heart disease, exercise training, GPAQ, exercise testing, strength testing

## Abstract

Although the efficacy of cardiac rehabilitation (CR) is proven, the need to improve patients’ adherence has emerged. There are only a few studies that have investigated the effect of sending text messages after a CR period to stimulate subjects’ ongoing engagement in regular physical activity (PA). A randomized controlled pilot trial was conducted after CR, sending a daily PA text message reminder to an intervention group (IG), which was compared with a usual care control group (CG) during three months of follow-up. Thirty-two subjects were assessed pre- and post-study intervention with GPAQ, submaximal iso-watt exercise testing, a 30 s sit-to-stand test, a bilateral arm curl test, and a final survey on a seven-point Likert scale. A statistically significant difference in the increase of moderate PA time (Δ 244.7 (95% CI 189.1, 300.4) minutes, *p* < 0.001) and in the reduction of sedentary behavior time (Δ −77.5 (95% CI 104.9, −50.1) minutes, *p* = 0.004) was shown when the IG was compared with the CG. This was associated with an improvement in heart rate, blood pressure, and patients’ Borg rating on the category ratio scale 10 (CR10) in iso-watt exercise testing (all *p* < 0.05). Furthermore, only the IG did not show a worsening of the strength parameters in the follow-up leading to a change of the 30 s sit-to-stand test with a difference of +2.2 (95% CI 1.23, 3.17) repetitions compared to CG (*p* = 0.03). The telemedical intervention has been appreciated by the IG, whose willingness to continue with regular PA emerged to be superior compared to the CG. Text messages are an effective and inexpensive adjuvant after phase 2 CR that improves adherence to regular PA. Further studies are needed to confirm these results in a larger patient population and in the long term.

## 1. Introduction

Cardiovascular diseases (CVD) are the major cause of mortality throughout the world. During the last decades, the mortality rate from CVD decreased, largely due to the increase in primary and secondary prevention strategies [1,2,3]. 

Secondary prevention focuses on controlling CVD risk factors, on reducing the impact of the disease by early diagnosis, and on developing a fast and effective treatment approach, which might be obtained by medical therapies and lifestyle interventions. Cardiac rehabilitation (CR) is a key component of secondary prevention in CVD and is a class IA recommendation in European guidelines [4]. CR should provide risk factor management, including psychosocial interventions, nutritional counseling, and physical activity (PA) prescription. CR programs are known to improve risk factor control and therapy adherence, increasing the quality of life and decreasing the recurrence of adverse events [5]. Patients who complete phase 2 CR are often encouraged to participate in phase 3 CR, whose purpose is to assist patients with long-term adherence to maintain regular exercise and other positive behavioral changes previously initiated [6]. 

Nevertheless, many patients do not comply with the current guidelines for secondary prevention and maintain smoking habits, an unhealthy diet, and a sedentary lifestyle. Participation and adherence to CR programs significantly decrease with age, socioeconomic status, barriers to access facilities, or concerns about exercise, as shown in Ruano-Ravina’s systematic review [7]. The rate of maintenance of regular PA is higher in patients who have participated in a CR program than in those who have not. Anyway, long-term adherence to PA after CR is still limited, being one of the main issues to improve in the near future [8].

Since 1970, telephone interventions have been used in cardiological counseling to encourage individuals to engage in physical exercise [9]. The number of these studies has increased significantly over the years but despite this, there is currently uncertainty regarding the effectiveness of these interventions [10,11]. Indeed, studies evaluating adherence to PA programs post-CR showed conflicting outcomes: on the one hand, improvements in exercise goals, blood pressure target, and adherence to medical therapy have been shown; on the other hand, there was no difference in smoking cessation, ability to meet low-density lipoprotein cholesterol targets, and hospital readmissions [11], resulting in low-quality evidence of mobile phone-based interventions [12,13,14,15]. 

Mobile health (mHealth) consists in the provision of medical assistance services through mobile communication devices. The most widespread mHealth solutions for secondary prevention were made available through smartphone applications. These have been demonstrated as a very promising tool to increase adherence to medical therapy, exercise prescriptions, and a healthy diet [16].

Digital health interventions could become a useful tool for healthcare professionals and patients after CR. The development of innovative, user-friendly, and cost-effective CR software programs has increased considerably during the last years. The real challenge today seems to be determining how to best integrate these new technological possibilities within pre-existing healthcare systems and outpatient clinics.

To the best of our knowledge, only a few studies have investigated the effect of sending text messages after phase 2 CR to incentivize subjects to engage in ongoing regular PA programs [17,18,19,20,21,22,23]. Furthermore, the design of the studies varied greatly and none of them included a specific PA protocol, and addressed the maintenance of the strength activity during the follow-up with mHealth initiatives. The relevance of strength training in the exercise prescription for cardiovascular patients is emphasized by the international guidelines [24], which state how specific training can improve muscular function as well as prevent the risk of falling and musculoskeletal injuries. Thus, the aim of this study was to investigate whether a standardized and inexpensive text messaging mHealth intervention post CR may improve patients’ adherence to PA and thereby exercise tolerance and muscle strength.

## 2. Materials and Methods

### 2.1. Setting and Participants

This study was carried out within the Rehabilitation Cardiology Unit of the Cittadella hospital (Veneto region, Italy) from September 2018 to July 2019. Participants were recruited from the patients admitted to a phase 2 CR program, lasting two weeks after acute ischemic heart disease. The patients started rehabilitation three weeks after percutaneous coronary intervention (PCI) or six weeks after coronary artery bypass grafting (CABG). Inclusion criteria were completion of the phase 2 CR program and possession as well as utilization of a mobile phone. The e criterion was a variation of the drug therapy during the three months post CR. 

### 2.2. Design

This study was a prospective, randomized controlled pilot trial on patients with ischemic heart disease during their follow-up after phase 2 CR. Each study participant received a written exercise prescription for the follow-up period, which included a warm-up phase, an aerobic component, a strength component, a balance training component, and a cool-down period. The recommended exercise frequency after CR was up to 5 times/week for the aerobic training and up to 3 times/week for the strength training. The training intensity was based on patients’ Borg rating on the category ratio scale 10 (CR10) or on the percentage of the maximal heart rate achieved in the maximal exercise test during CR. All patients were informed about the study protocol and provided written informed consent to participation. Moreover, this study analyzed data of patients’ therapeutic pathway of routine clinical assistance and all assessments were part of a standardized clinical approach. An external subject performed a blinded block randomization to ensure a similar number of patients between the two study groups, i.e., the intervention group (IG) and control group (CG). 

### 2.3. Intervention

The study intervention that was provided to the IG for three months during follow-up after CR consisted in the following standardized text message: “The rehabilitation cardiology service reminds you to carry on with your PA program as indicated in the prescription.” The daily recall, scheduled at 8.30 a.m., was sent to all participants of the IG using the smartphone-app “Do It Later”, encouraging them to perform the assigned exercise prescription. Instead, patients in the CG did not receive any text message during the three-month follow-up. However, an exercise prescription was provided after phase 2 CR, which was recommended to be performed during these months.

### 2.4. Endpoints and Assessments

All evaluations were performed after phase 2 CR and repeated at the three-month follow-up investigation. The primary endpoint of the study was to evaluate the effect of a daily text message recall on maintaining good PA habits as determined by the Global Physical Activity Questionnaire (GPAQ) version 2. The GPAQ was administered by a trained, in-person interviewer.

Secondary outcome measures included physical performance and exercise tolerance that were evaluated through a submaximal iso-watt exercise test before and after the follow-up period with study intervention. Each patient’s iso-watt intensity was identified as the power (watts) achieved and maintained for a duration of 5 min at a heart rate between 55% and 65% of the heart rate reserve, using the Karvonen method. For the determination of each patient’s maximal HR, an incremental maximal cycle ergometer stress test was used, performed during the CR program. The submaximal iso-watt exercise test lasted 15 min, including 10 min of incremental warm-up and 5 min of iso-watt exercise. The following parameters were measured: (i) resting heart rate (HR); (ii) resting blood pressure (BP); (iii) CR10 Borg rating reached during submaximal effort; (iv–v) mean HR and BP during the 5 min iso-watt exercise. Moreover, strength parameters were evaluated with a 30 s sit-to-stand test and a bilateral arm curl test. Lastly, at the three-month follow-up evaluation, a survey on a seven-point Likert scale was provided to all the participants, evaluating the following items: (i) the usability of the heart rate monitor; (ii) the compliance with the exercise prescription; (iii) the satisfaction with the rehabilitation and follow-up services offered; (iv) the willingness to continue with regular PA in the future; (v–vi) the impact of the CR and the mHealth recall on the subject’s psycho-physical state. Finally, the occurrence of cardiovascular symptoms and the number of hospitalizations during the follow-up were investigated.

### 2.5. Statistical Analyses

The sample size was determined considering a statistical power of 80% to identify the difference with a significance of 0.05 [25]. Moreover, the mean value and standard deviation of the moderate recreational activity was evaluated by the GPAQ, previously measured in a sample of subjects with similar clinical conditions before starting cardiac rehabilitation (160 ± 150 min/week) and an expected average mean value of 310 min/week after intervention in the IG was considered for power calculation. This value was expected by estimating an average increase of 150 min/week with respect to basal moderate PA levels. Indeed, this value could guarantee that at least 85% of the subjects in the IG could reach the amount of 150 min of moderate physical activity per week. Thus, calculating a possible dropout rate of 10%, it was aimed to include 36 participants in this study.

The Shapiro-Wilk test was used to verify whether the variables under consideration had a normal distribution. Continuous variables were expressed as mean (±standard deviation) and the comparison between subgroups was performed with Student’s *t*-test or the Wilcoxon-Mann-Whitney test. Categorical variables were expressed as frequencies and percentages and were compared between groups using Pearson’s chi-square test. The analysis was based on the intention-to-treat principle, evaluating all participants according to the initial randomized assignments.

## 3. Results

### 3.1. Characteristics of the Study Population

After four dropouts before the randomization due to intercurrent diseases, 32 patients were included in the study (age 61.2 ± 9.0 years, 85% male) and randomly assigned to the IG or CG, respectively (Figure 1). No significant differences emerged between the IG and CG with regard to demographic characteristics, coronary artery disease treatment (PTCA or CABG) and comorbidities (Table 1). 

### 3.2. GPAQ

The GPAQ analysis, which evaluated the difference in PA levels between baseline and after three months of follow-up, demonstrated a statistically significant difference in the increase of moderate PA time (Δ 244.7 (95% CI 189.1, 300.4) minutes, *p* < 0.001) and in the reduction of sedentary behavior time (Δ −77.5 (95% CI −104.9, −50.1) minutes, *p* = 0.004) in favor of the IG compared to the CG. GPAQ data are presented in Table 2 and Figure 2. 

### 3.3. Submaximal Iso-Watt Exercise Test 

The work intensity of the performed submaximal iso-watt exercise test was similar in both groups (IG: 52.2 +/− 15.4 vs. CG: 51.3 +/− 10.9 watt; *p* = 0.84). No minor or severe adverse events were registered during the submaximal exercise tests. 

No statistically significant differences emerged between the baseline evaluation and the three-month follow-up assessment regarding the patients’ resting HR, resting systolic BP (SBP) and resting diastolic BP (DBP) in both the IG and CG.

A statistically significant decrease of iso-watt HR, SBP and of CR10 rating of perceived exertion was observed in the IG group after follow-up, which leads to a significant difference compared to the CG (Table 3).

### 3.4. Strength Tests

No statistically significant differences emerged between the baseline evaluation and the three-month follow-up assessment in bilateral arm curl test, comparing the IG with the CG. However, a statistically significant difference emerged in the comparison between both study groups in favor of those patients regularly contacted by text messages, showing a better outcome on the 30 s sit-to-stand test (Δ +2.2 (95% CI 1.23, 3.17) repetitions, *p* = 0.03). Strength test results are presented in Table 4.

### 3.5. Final Follow-Up Survey

Patients in both groups demonstrated a good and similar compliance with the use of the HR monitor (7/7 on a Likert scale—“always”, *p* = 0.92) and complied with the HR target during exercise (7/7—“always”, *p* > 0.99). The majority (53.3%) of the IG patients rated the daily text message as a “very useful” incentive (7/7) and were “moderately satisfied” (6/7) with the text message method. Interestingly, at the question “Will I engage in regular PA in the future?” the IG appeared more motivated to continue with PA compared with the CG (IG rated 7/7, corresponding to “completely true”; CG rated 6/7, corresponding to “partially true”; *p* = 0.04). Concerning subjects’ mental and physical state, most of the responses reported it as “really improved” (7/7), independently of group in the study. Finally, at the three-month follow-up survey, no re-hospitalization or major exercise limiting signs or symptoms emerged in the entire study population.

## 4. Discussion

This study highlights a preliminary efficacy of mHealth applications such as text messages as a daily reminder to improve adherence to adequate exercise and PA levels in the post-CR setting. The results showed a significant decrease in heart rate, blood pressure and CR10 Borg rating at iso-watt exercise testing, in favor of the IG. This improvement in submaximal exercise tolerance was associated with a significant amelioration in the level of PA for the IG compared with the CG. In particular, the GPAQ demonstrated an increase in moderate recreational physical activity and a reduction in sedentary behavior, thus confirming previous data [19,20,21,22]. 

These higher levels of PA and the associated beneficial effects of the study intervention on submaximal work capacity could positively affect the activities of daily living [26]. In accordance with previous studies, no significant changes appeared in HR, SBP, and DBP at rest in both the IG and CG, respectively [17,20,21,22]. Only Chow et al. in 2015 demonstrated a lowering of resting SPB after a longer, six months, follow-up period [23]. However, exercise test parameters demonstrated the effectiveness of the intervention in improving exercise tolerance as shown by a reduction of HR, SBP and Borg ratings of perceived exertion at submaximal exercise intensities. These training adaptations were not reached in the CG. These objectively measured exercise test outcomes may further strengthen what has emerged from subjective reports of the GPAQ, i.e., that mHealth applications might be useful in improving the adherence to PA after CR, leading to better physical fitness and thus a lower CV risk [27].

The present study is noteworthy for the implementation of a standardized adapted aerobic and strength exercise prescription after CR with the aim to maintain or improve also the patients’ muscular strength during follow-up. Despite the prescribed strength training exercises, both IG and CG worsened bilateral arm curl strength at three months of follow-up. These results may suggest the scarce compliance with non-supervised physical exercise, or the fear of possible adverse effects related to upper-body strength exercises [28], especially for those who have undergone a sternotomy. Nevertheless, the IG showed a better outcome on the lower-body strength, which has been maintained during the follow-up period in the IG, while it decreased in the CG. This positive impact could be achieved with just simple text messaging. These data highlight the need for further implementing the prescription and explaining the benefits and safety of strength exercises [29].

In addition, the telemedical recall intervention appeared to be feasible and effective, with a greater impact on improving compliance and patients’ motivation [18,23]. Moreover, the beneficial effects of regular exercise on mental and physical health are known and patients reported improved well-being at the three-month follow-up evaluation [21]. Moreover, as previously reported by other studies, the mHealth intervention has been appreciated by all patients [22,23], and was associated with a stronger motivation to continue with regular exercise in the near future compared to the CG. This may be due to the fact that the IG got more benefits from PA and received regular mental incentives through text messaging. The effectiveness and safety of this CR follow-up intervention was also shown by the absence of rehospitalizations, minor or severe adverse events or symptoms during the three months of home-based exercise training [20]. This might have been also safeguarded by the individualized exercise prescription and the patients’ good compliance in the use of the HR monitor. Indeed, the study participants indicated to have controlled and respected the target HR during the exercise training sessions.

Finally, the present study further corroborates the safety and possible efficacy of text messages as mHealth application to increase exercise adherence in the follow-up after phase 2 CR and suggests the need for further implementation of strength training prescription. Especially for those who cannot participate in a structured exercise program supervised by exercise professionals, periodic reminders with text messages or other mHealth methods might be useful to introduce [30]. Periodic telephone interviews to adjust the exercise prescription could also be an additional resource to improve the efficacy of CR. 

### Limitations and Perspectives

Although maximal cardiopulmonary exercise testing is the gold standard evaluation to measure patients’ peak aerobic capacity and fitness, this study reports data on submaximal exercise capacity, which reflects functional work capacity during the activities of daily living and may thus affect quality of life. Furthermore, the follow-up period lasted only three months, sending a general text message to the IG. Thus, future studies are needed to confirm these outcomes in the long term and should also investigate the impact of a more adapted text message to patients in order to further increase their motivation and adherence to exercise. The study conclusions might be limited by the restricted sample size, but the outcomes may provide a realistic depiction of the utility of text messaging in the usual care. Although this pilot study comes with some limitations, it specifically investigates the impact of a simple and standardized mHealth text message intervention in a real-world CR setting with a randomized controlled approach. Moreover, a specific exercise prescription has been provided and strength training has been explicitly addressed, thereby providing new perspectives for clinical practice and future trials.

## 5. Conclusions

Sending standardized telephone text messages, as an add-on mHealth application to individualized exercise prescription, was shown to reinforce the outcomes after a phase 2 CR program. Indeed, it is a safe, effective, and inexpensive intervention that improves adherence to good PA habits in patients after cardiac revascularization procedures and leads thereby to improved exercise tolerance. Moreover, strength training should be specifically addressed during patients’ follow-up after CR, as indicated by international guidelines. This feasible mHealth intervention is well accepted by patients who have reported higher motivation to engage in regular PA also in the next future. Future research projects on phase 3 CR and the adherence to regular exercise could focus on a more specific strength training intervention and tailored text messaging, to investigate whether a structured mHealth intervention can confirm these results also in the long term.

## Figures and Tables

**Figure 1 ijerph-18-06645-f001:**
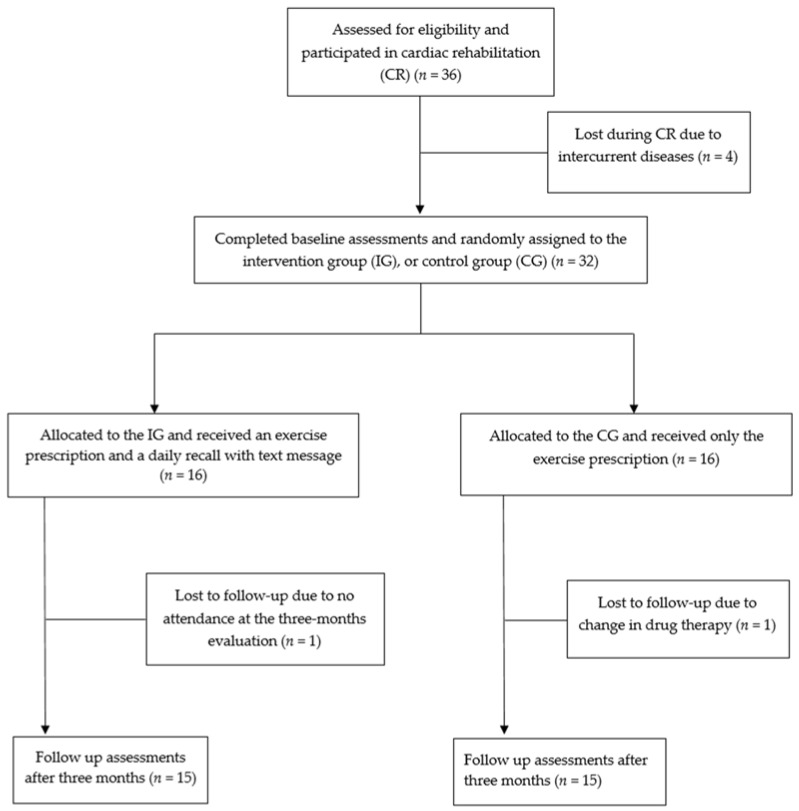
PRISMA flow diagram.

**Figure 2 ijerph-18-06645-f002:**
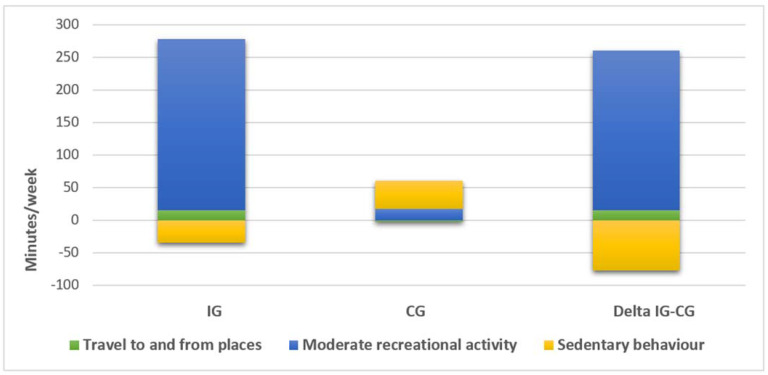
Patients’ modifications in physical activity after cardiac rehabilitation. Variations in physical activity levels evaluated with GPAQ in the intervention group (IG) and the control group (CG). The differences between the three-month follow-up and the baseline assessments are shown for both study groups. Furthermore, a comparison between the two groups is illustrated (Δ IG-CG).

**Table 1 ijerph-18-06645-t001:** Baseline characteristics of the study population.

	Control Group (*n* = 16)	Intervention Group (*n* = 16)
Age (DS) years	61.1 (10.6)	61.4 (8.9)
Men (%)	13 (81.3)	14 (87.5)
BMI (DS) kg/m^2^	27.0 (2.9)	26.6 (3.2)
Cardiac pathology *n* (%)		
HFrEF	1 (6.3)	1 (6.3)
STEMI	5 (31.3)	5 (31.3)
NSTEMI	3 (18.8)	2 (12.5)
Type of intervention *n* (%)		
PTCA	12 (75.0)	12 (75.0)
CABG	4 (25.0)	4 (25.0)
Comorbidities *n* (%)		
Dyslipidemia	12 (75.0)	11 (68.8)
Arterial hypertension	10 (62.5)	9 (56.3)
Chronic kidney disease	1 (6.3)	3 (18.8)
Impaired fasting glucose	4 (25.0)	4 (25.0)
Type 2 diabetes mellitus	2 (12.5)	3 (18.8)

Baseline characteristics of the study population. No significant difference exists between the two groups. HFrEF: heart failure with reduced ejection fraction; STEMI: ST segment elevation myocardial infarction; NSTEMI: non-ST segment elevation myocardial infarction; PTCA: percutaneous transluminal coronary angioplasty; CABG: coronary artery bypass graft. Data are presented as mean (±standard deviation) or as number (%).

**Table 2 ijerph-18-06645-t002:** Patients’ physical activity level.

	Control Group (CG)		Intervention Group (IG)		Δ between Groups
	Baseline	Three Months	Baseline	Three Months	Δ IG (3 Months—Baseline)–Δ CG (3 Months—Baseline)
Travel to and from places (min)	43.8 (89.6)	43.4 (68.2)	134.6 (319.0)	150.0 (316.0) *	15.3 (−13.7, 44.3)
Moderate recreational activity (min)	150.0 (98.4)	166.9 (158.5)	119.4 (111.4)	380.9 (167.4) ***	244.7 (189.1, 300.4) ***
Sedentary behavior (min)	251.3 (88.5)	294.4 (44.1) *	255.0 (91.7)	220.6 (89.5)	−77.5 (−104.9, −50.1) **

GPAQ questionnaire results at baseline and at three-month follow-up in both study groups. Data are presented as mean (±standard deviation), except for the column of “Δ between groups”, where data are presented as mean (95% confidence interval). Asterisks identify the presence of a statistically significant difference between three-month follow-up and basal conditions. * *p* < 0.05; ** *p* < 0.01; *** *p* < 0.001.

**Table 3 ijerph-18-06645-t003:** Patients’ performance at submaximal exercise testing.

	Control Group (CG)		Intervention Group (IG)		Δ between Groups
	Baseline	Three Months	Baseline	Three Months	Δ IG (3 Months—Baseline)–CG (3 Months—Baseline)
Resting HR (bpm)	62.7 (6.4)	63.1 (6.3)	63.9 (4.2)	62.6 (5.2)	−1.7 (−3.9, −0.3)
Resting SBP (mmHg)	120.6 (12.6)	121.1 (13.3)	122.1 (7.4)	120.5 (8.9)	−2.1 (−4.6, 0.4)
Resting DBP (mmHg)	72.0 (7.9)	73.1 (6.8)	74.4 (7.7)	74.1 (8.0)	−0.6 (−1.5, 0.3)
Exercise HR (bpm)	87.5 (8.9)	91.8 (9.6) *	90.7 (8.7)	88.5 (9.8) *	−6.6 (−9.5, −3.7) **
Exercise SBP (mmHg)	148.3 (17.5)	153.1 (19.5) *	149.2 (15.4)	144.4 (16.8) *	−9.6 (−12.5, −6.7) **
Exercise DBP (mmHg)	76.3 (9.0)	80.3 (10.2) *	77.2 (8.0)	76.3 (7.2)	−5.0 (−7.1, −2.9) *
CR10	3.9 (0.9)	4.6 (1.5) *	4.1 (1.3)	3.6 (1.7) *	−1.2 (−1.7, −0.8) **

Resting cardiovascular data and response to exercise evaluated during a submaximal iso-watt exercise test before and after a three-month follow-up period post CR. HR: heart rate SBP: systolic blood pressure; DBP: diastolic blood pressure; CR10: patients’ Borg rating on the category ratio scale 10. Data are presented as mean (±standard deviation), except for the column of “Δ between groups”, where data are presented as mean (95% confidence interval). Asterisks identify the presence of a statistically significant difference between three-month follow-up and basal conditions within each group or between the two groups. * *p* < 0.05; ** *p* < 0.01.

**Table 4 ijerph-18-06645-t004:** Patients’ performance at strength testing.

	Control Group (CG)		Intervention Group (IG)		Δ between Groups
	Baseline	Three Months	Baseline	Three Months	Δ IG (3 Months—Baseline)–Δ CG (3 Months—Baseline)
30 s chair sit-to-stand test (reps)	14.1 (3.4)	12.3 (3.2) **	15.6 (3.8)	16.0 (4.7)	2.2 (1.23, 3.17) *
Right arm curl test (reps)	14.7 (3.8)	12.8 (4.6) *	15.8 (4.2)	14.0 (3.9) *	0.1 (−0.59, 0.79)
Left arm curl test (reps)	13.8 (3.9)	12.0 (4.6) **	14.3 (3.0)	12.6 (3.4) **	0.0 (−0.52, 0.52)

Strength test results at baseline and at three-month follow-up in both study groups. Reps: repetitions. Data are presented as mean (±standard deviation), except for the column “Δ between groups”, where data are presented as mean (95% confidence interval). Asterisks identify the presence of a statistically significant difference between three-month follow-up and basal conditions or between the two study groups. * *p* < 0.05; ** *p* < 0.01.

## Data Availability

The data presented in this study are available on request from the corresponding author.

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
