# Peer review of "Effectiveness of Text Messaging as an Incentive to Maintain Physical Activity after Cardiac Rehabilitation: A Randomized Controlled Pilot Study"

_ijerph, 2021, doi:10.3390/ijerph18126645_

Round 1

Reviewer 1 Report

The aim of this study was to investigate whether a text-messaging mHealth intervention post cardiac rehabilitation may improve patients’ long-term adherence to physical activity, exercise tolerance and muscle strength. The conclusions obtained indicate that these messages could improve the adherence in terms of physical activity and exercise tolerance, although there is scarce benefit over muscle strength.

The objective pursued is interesting, and I agree with all your work’s highlights. However, I have observed various aspects that should be clarified, in order to achieve a better comprehension of the manuscript. I apologize if any aspect of the manuscript has been misunderstood, and I encourage to the authors to argue it.

MAJOR REVISION

ABSTRACT

P-values must be cited in the abstract.

L.20: what did the authors understand as “usual follow-up”?

INTRODUCTION

In my opinion, phase-3 of cardiac rehabilitation should be cited, as it is a way to improve patients’ adherence.

MATERIAL AND METHODS

Design, L.78: “admitted to a two-weeks phase II – CR program”. I’m not sure if authors are saying that this phase II has a total duration of two weeks.

Design: All patients where ischemic heart disease, but I would ask to authors if they have the same risk, in terms of cardiopulmonary testing or ejection fraction, for example.

Intervention: a three-month follow-up after cardiac rehabilitation is very short time. The aim of your study is investigate if a mHealth intervention post cardiac rehabilitation could improve patients’ long-term adherence to physical activity, exercise tolerance and muscle strength. Three months is not a long-term follow-up.

Intervention: from my point of view, this text-message is too much general. With a sample size of 32 patients, I think that these messages should be more adapted to the patients, or even change throughout the days, to maximize patients’ motivation.

Endpoints and assessments: why have you chosen a submaximal iso-Watt exercise test instead of a maximal Cardiopulmonary Exercise Testing? This CPET is the Gold Standard for patients with ischemic heart disease.

RESULTS

I don’t see Figure 1 in text.

DISCUSSION

Limitations section should be longer. Authors haven’t said anything about their short follow-up, for example.

I think that conclusion (or even discussion) should include proposals for future research.

MINOR REVISION

INTRODUCTION

I think that IJERPH instructions for authors say that all references should be placed in square brackets, and placed before the punctuation.

DISCUSSION

L.210: “Authors”. This word doesn’t fit in the sentence.

Reviewer 2 Report

This paper tests a simple text messaging system used to improve PA after cardiac rehabilitation. Significant changes are required throughout the manuscript to ensure clarity of results and recognize limitations of the study.

Title: include study design in the title

Abstract:

As this is only a small trial, please describe as a pilot RCT

Describe the outcome measures used including those related to patient satisfaction

Report between group findings only and include mean differences with CI's

Introduction

Please avoid use of conjunctive adverbs throughout the manuscript. They distract from the main message.

Please provide more explanation about why people don't comply with guidelines

Please expand on the limited adherence to PA guidelines after CR e.g. how many people are meeting guidelines

Change structure of the intro to introduce the telephone interventions section before talking about more advanced telehealth and describe in more detail about what existing studies related to text messaging and CR are to really highlight the gap. You also state strength training is important - expand why. Finish this section with line 52-54

You say you are testing a feasible text messaging service. How do you know it is feasible? has it previously been tested - if so provide this as background, if not, delete.

Methods:

Please clarify whether recommendation for 5xweekly training is during or after CR.

line 90 replace strength part with strength training

Please explain the relevance of analysing the clinical pathway and include somewhere in the results

Statistical analysis

Using ANCOVA would be more appropriate to account for baseline differences. There seems a large difference in baseline PA which is your primary outcome

Please state if there was intention to treat

Results

Add a PRISMA flow diagram. Were the dropouts before or after randomization?

Throughout the results do not report within group changes as this was not the research question. Only report between group changes and include MD and confidence intervals

Column 3 in table 2 does not seem to accurately reflect 3 month change between groups e.g. 150-43.4 = 106.6 not 15.3?

Ensure same decimal place is reported throughout

Discussion

As this is a small trial reduce the emphasis on effectiveness with ‘preliminary efficacy’ and state what outcomes showed improvement

Remove paragraphs about the efficacy of cardiac rehab – this study is about text messaging not the results from the CR program

Suggest restructuring to

  1. Talk about improvements in PA and how this fits with current evidence
  2. Feasibility and patient satisfaction of using text messages in clinic
  3. Why it might have been effective
  4. Safety of text messaging

Include strengths and limitations section – what reporting checklist did you use ? acknowledge it is a small trial . Was there any blinding? All could introduce potential bias

Round 2

Reviewer 1 Report

Thank you for your response, that has clarified mostle of the aspects that I have asked. 

Reviewer 2 Report

Thank you for making suggested changes however I still have some concerns before the paper can be published mainly about the presentation of the results.

The way the results are currently written is still confusing. In some sections e.g. line 187 section 3.2 and section 3.4 line 221 you talk about differences between baseline and 3 month followup which detracts from the research question. Include the baseline resultsfor GPAQ and strength in the table  so no further explanation is required. The results section should be consistent between sections as is layed out in section 3.3 with a focus on between group differences only.

Following on from this point, the second paragraph should again focus on the outcome of the main research question. Even though this is an interesting point that strength training reduced in both groups, it is not the main point. Move this point later in the discussion and focus on the between group difference in strength ie IG group had indeed better strength. Therefore you could still argue the point but be more sussinct and direct with the main finding ie the intervention group had better strength at 3 months. This is noteworthy given both groups reduced their strength over all - then state explanation why.

Line 243 change to this study highlights the preliminary efficacy

It is important to also outline strengths as well as limitations. Even though this journal did not request a reporting checklist, it will make your research more robust to ensure you have addressed all the requirements required of reporting an RCT. As you  state you have done this, it should be easy to go back through the manuscript and match it with the CONSORT checklist for RCT's
